# Dendrimers as Pharmaceutical Excipients: Synthesis, Properties, Toxicity and Biomedical Applications

**DOI:** 10.3390/ma13010065

**Published:** 2019-12-21

**Authors:** Ana Santos, Francisco Veiga, Ana Figueiras

**Affiliations:** 1Department of Pharmaceutical Technology, Faculty of Pharmacy, University of Coimbra, 3004-531 Coimbra, Portugal; anaicfsantos@gmail.com (A.S.); fveiga@ff.uc.pt (F.V.); 2REQUIMTE/LAQV, Group of Pharmaceutical Technology, Faculty of Pharmacy, University of Coimbra, 3004-531 Coimbra, Portugal

**Keywords:** dendrimers, pharmaceutical excipient, drug-delivery systems, toxicity, physicochemical properties, synthesis, biodistribution

## Abstract

The European Medicines Agency (EMA) and the Current Good Manufacturing Practices (cGMP) in the United States of America, define excipient as the constituents of the pharmaceutical form other than the active ingredient, i.e., any component that is intended to furnish pharmacological activity. Although dendrimers do not have a pharmacopoeia monograph and, therefore, cannot be recognized as a pharmaceutical excipient, these nanostructures have received enormous attention from researchers. Due to their unique properties, like the nanoscale uniform size, a high degree of branching, polyvalency, aqueous solubility, internal cavities, and biocompatibility, dendrimers are ideal as active excipients, enhancing the solubility of poorly water-soluble drugs. The fact that the dendrimer’s properties are controllable during their synthesis render them promising agents for drug-delivery applications in several pharmaceutical formulations. Additionally, dendrimers can be used for reducing the drug toxicity and for the enhancement of the drug efficacy. This review aims to discuss the properties that turn dendrimers into pharmaceutical excipients and their potential applications in the pharmaceutical and biomedical fields.

## 1. Introduction

Nanotechnology is an emergent area that studies materials with a nanometer-scale [1]. During recent decades, nanotechnology has received great interest from researchers in the fields of biomedical engineering, pharmaceutical technology, and medicine. Nanomaterials (NM) are structures with less than 100 nm in at least one of their dimensions that have unique physical, chemical, and/or biological properties associated with their nanostructure [2]. Nanomedicine is one of the sub-topics of nanotechnology, having as its main purpose the treatment and prevention of diseases through nanoformulation [3,4]. One of the major goals of nanotechnology and nanomedicine is to develop a good pharmaceutical formulation, i.e., to produce a safe and effective drug formulation, with quality, while enhancing the bioavailability of the active pharmaceutical ingredients (APIs).

Some APIs have inherent bioavailability due to their good solubility and permeation through biological membranes. However, many of them belong to class II of the Biopharmaceutics Classification System (BCS) (i.e., low solubility, high permeability) or to class IV (low solubility, low permeability), which is translated into low bioavailability [5]. Almost 40% of the APIs developed by the pharmaceutical industry are rejected due to bioavailability problems [6]. One way to overcome the bioavailability drawbacks of the API is by choosing appropriate excipients for the pharmaceutical formulation, improving the dissolution profile of the drug.

The European Medicines Agency (EMA) “Guideline on excipients in the dossier for application for marketing authorization of a medicinal product” [7] and the Current Good Manufacturing Practices (cGMP) [8] define an excipient or inactive ingredient as the constituents of the pharmaceutical form other than the active ingredient (i.e., “any component that is intended to furnish pharmacological activity or other direct effect in the diagnosis, cure, mitigation, treatment, or prevention of disease, or to affect the structure or any function of the body of human or other animals”). The European Commission guideline on “Excipients in the labelling and package leaflet of medicinal products for human use” [9] defines excipients as “any constituents of a medicinal product, other than the active substance and the packaging material”. This guideline recognizes that, although excipients are usually inert and with little or no pharmacological effect, some may have some action in certain circumstances.

Due to the unique properties that NMs offer, they can be used to reduce some of the limitations found in traditional pharmaceutical formulations. In other words, these NMs may be the key to solving the bioavailability problems of the APIs referred above, being used as excipients.

Among the various NMs, dendrimers have been highlighted in recent years as promising nanostructures, and their applicability as pharmaceutical excipients has been explored, due to their distinctive physicochemical and structural properties [10]. Dendrimers are highly branched polymers with surfaces that are easily modifiable. In pharmaceutical technology, these polymers can be considered as excipients in the development of several pharmaceutical forms. Due to certain properties such as nanoscale size, a high degree of branching, polyvalency, biocompatibility, high water solubility, absence of immunogenicity, precise molecular weight, and available internal cavities, dendrimers are excellent vehicles for safely and effectively transporting drugs [11]. Additionally, the possibility of combining different polymeric excipients, such as polyethylene glycol (PEG), enables the production of controlled drug-delivery systems and the development of novel pharmaceutical formulations [12].

The unique properties of dendrimers, which differentiate them from other NMs, render them as widely applicable in diagnostic and biomedical engineering, including drug and gene delivery systems. The use of dendrimers for drug targeting and delivery has proven to have an important role in improving drug safety and reducing drug-related toxicity [13].

Another interesting application of dendrimers is in inflammatory diseases because these nanostructures present anti-inflammatory activity by themselves, being useful in the treatment of atherosclerosis, rheumatoid arthritis, and other associated diseases [13,14]. Additionally, dendrimers may also have antimicrobial and antiviral activities [13,15].

Although the properties of dendrimers render them suitable as pharmaceutical excipients, they also present less advantageous properties, which may hinder their use, namely, their cytotoxicity, the limitation of incorporation of the drug into the dendrimer cavities, and the inability to control the rate of drug release. Furthermore, this type of polymer also presents high manufacturing costs and the need for a specialized workforce [13,16]. However, some of these problems are nowadays less significant due to the evolution of nanotechnology and increasing knowledge about dendrimers. These developments have allowed for increased industrial manufacturing efficiency and lowered production costs [16].

In order to reduce the cytotoxicity and to increase the space on the dendrimer cavity, numerous modifications have been proposed to the chemical structure of the dendrimers. Several studies have shown that cytotoxicity of dendrimers could be significantly reduced by surface modifications with inert particles such as PEG [17] and fatty acids [13]. These alterations make dendrimers more suitable for use as pharmaceutical excipients. Additionally, PEGylation of dendrimers increases their blood circulation time [17]. The lack of control of the rate of drug release from the dendrimer can be avoided by covalent conjugation of the drug to the dendrimer surface. Drug release is then dependent on the cleavage of the dendrimer–drug linkage [13].

## 2. Dendrimers

### 2.1. Definition and Structure

In 1985, the synthesis of “nanocascade spheres” and “starburst dendritic macromolecules” was reported, introducing a new class of macromolecules, known today as dendrimers [18]. The word dendrimer has its routes from the Greek word *dendron*, meaning “tree” or “branch”, and the word *meros*, meaning “part” [19]. Dendrimers, or dendritic polymers, are nanoparticles (between 1 nm to 100 nm of diameter), and sometimes, they can be used as drug-delivery systems, [6,20]. These nanodelivery platforms can increase drug solubility and, subsequently, improve drug bioavailability. Additionally, they can reduce the therapeutic dosage by increasing the drug time of exposure, minimizing adverse drug side effects [21].

Dendrimers are three-dimensional, hyper-branched and monodisperse structures containing a central core surrounded by peripheral groups. These characteristics are fundamental for their physicochemical and biological properties. Normally, dendrimers possess three distinguishing architectural components, as represented in Figure 1: (i) core; (ii) branches (an interior layer composed of repeating units attached to the core) and (iii) terminal groups attached to the branches [18]. The dendritic polymer arrangement creates internal cavities, in which the drug can be deposited, increasing its solubility and stability. The referred characteristics turn these macromolecules into good candidates for pharmaceutical excipients [19,22].

The core of dendrimers consists of an atom, or group of atoms, with branches of carbon that are added through a sequence of chemical reactions, producing a spherical dendritic structure.

Dendritic polymers are synthesized stepwise around the core. During the synthesis, each successive reaction step leads to an additional generation of branching, and the number of repeated steps is defined as a dendrimer generation (denoted as G), as represented in Figure 2. Normally, dendrimers with G < 4 and G ≥ 4 are termed low- and high-generation dendrimers, respectively [23]. The core of the dendrimer is sometimes denoted as generation zero or G0 [24].

The terminal groups (also called functional groups or surface groups) are responsible for the interaction of the dendrimers with the external groups or molecules. Consequently, the physicochemical properties of dendrimers depend not only on the branching units, but also on the surface functional groups [25]. As the control of the physicochemical properties of these structures can be achieved during synthesis by controlling the core groups, the extent of branching, and the nature and/or number of terminal groups on the surface, dendrimers became popular in the NMs research area [26]. This leads to the possibility of acquiring, at the end of the chemical synthesis, the desired chemical structure with an appropriate role: as API or as a pharmaceutical excipient (Figure 3).

When dendrimers act as APIs, the structure determines if they possess anti-inflammatory, antiviral or antimicrobial activity. Park et al. [28] studied the anti-inflammatory effect and reported that dendrimers with glucosamine conjugates inhibited the synthesis of pro-inflammatory chemokines and cytokines of cells associated with the inflammatory process (such as monocytes, macrophages, and lymphocytes). Their results demonstrate that such conjugates also block the proliferation and migration of endothelial cells mediated by a growth factor produced by fibroblasts.

Frequently, antiviral and antimicrobial treatment is hampered by drug resistance and the appearance of adverse effects, increasing the need for new therapeutic strategies. Dendrimers demonstrate good antiviral and antimicrobial activities, due to the strong interactions that they make with a virus and the bacterial membranes, respectively, preventing the infection of the host. Consequently, they became an important tool in the treatment of viral infections, especially in human immunodeficiency virus (HIV) and influenza virus infections [13]. An example of a dendrimer with antiviral action is the SPL7013 (VivaGel^®^) [29,30,31]. This nanostructure is a Poly-L-lysine (PLL) dendrimer which presents an anionic naphthalene disulphonate surface. They can block the entry of the virus by binding to the viral envelope protein gp120, preventing the formation of the CD4-gp120 complex [32]. They also exhibit potent activity against bacterial vaginosis when administered as a topical gel [33]. VivaGel^®^ was used as the first dendrimer-based commercial medical product and many clinical trials with dendrimers are now being conducted [25]. Nowadays, VivaGel^®^ is also used in condoms, already available on the market in Australia and Japan [34], with 99.9% effectiveness against HIV [33].

As excipients, dendrimers may be used to improve the physicochemical properties of a pharmaceutical formulation. In fact, the concept of excipients evolved: at first, excipients were used simply as substances added to complete a volume in the formulation; currently, these substances also serve as a vehicle and to incorporate the API. Excipients are classified as solubilizers, permeation enhancers, dyes, emulsifiers, diluents, flavors, preservatives, wetting agents, solvents, and sustained-release matrices [9]. In order to be good excipients, dendrimers should not alter the safety, efficacy and stability of the formulation, while guaranteeing that the dose is administered and delivered with precision and accuracy [35].

Besides poly(amidoamine) (PAMAM), there are other types of dendrimer structures that can be excipients, as listed in Table 1. These dendrimers share some common characteristics, namely biocompatibility, high aqueous and non-polar solubility, being monodisperse, and the non-existence of a linear relationship between viscosity and molecular weight [19].

Poly(amidoamine) (PAMAM) dendrimers are the best-studied and most used nanostructures. The structure and properties of PAMAM dendrimers render them suitable for the fixation and encapsulation of drugs and can be used to improve solubilization, residence time, bioavailability and permeation through the skin. Consequently, PAMAM dendrimers play an important role in various pharmaceutical applications, more particularly, in new ocular, pulmonary, transdermal, and oral drug-delivery systems [19,41]. Table 2 summarizes the effects of the use of PAMAM dendrimers in several pharmaceutical applications.

In 1991, a major breakthrough was made in the area of NMs with the discovery of a new class of dendrimers, known as Janus dendrimers (JDs) or diblock dendrimers [45] These dendrimers attracted much attention of researchers because of their asymmetric structure, which grants them unusual properties, when compared to the conventional symmetrical forms. Sometimes JDs are composed of two hemispheres (hydrophobic and hydrophilic) with different numbers of terminal groups and sizes. Diblock dendrimers play a significant role in medical and drug-delivery applications because of their capacity to increase drug solubility, the ability to effectively carry drugs and for their potential as transdermal penetration enhancers [46]. Ouyang et al. [47] and Pan et al. [48] studied the performance of the JDs for bone-targeted drug delivery of naproxen, a class II drug according to the BCS. They concluded that this amphiphilic dendrimer has the ability to enhance the water solubility, to carry the drug, and to target it to the bone. Therefore, this amphiphilic JD was revealed to be a promising approach to delivery of poor water-soluble drugs in specific targets. Another study was conducted by Liu et al. [49] that demonstrated that JDs act as a potential siRNA delivery system. They confirmed that these amphiphilic dendrimers have the ability to be a robust and versatile siRNA delivery system to various cell lines, including human primary and stem cells.

Although dendrimers do not have a pharmacopoeia monograph and cannot yet be recognized as a pharmaceutical excipient, significant progress has been made towards establishing dendrimers as pharmaceutical excipients [16].

### 2.2. Synthesis

Dendrimers’ synthesis is related to molecular and polymer chemistry. They associate with the molecular chemistry world by their step-by-step controlled synthesis, and they are related to the polymer world due to their use of repetitive structure, i.e., monomers. These globular structures are synthesized in cascade by a sequence of reactive steps to grow from the first generation (G1) to the second generation (G1 + 1), and so on [50].

The first synthesized dendrimers were PAMAMs, introduced in 1980. However, various other dendrimers including poly(propyleneimine) (PPI) and poly-L-lysine (PLL), glycodendrimers, polyester dendrimers, and amphiphilic dendrimers, were synthesized in later years [51].

#### 2.2.1. Classical Synthesis Pathways

Dendrimers are usually synthesized through methods that allow the control of the structure at every stage of construction. The dendritic structures are mostly synthesized by two main different methods: divergent or convergent [52].

##### Divergent Growth Method

The divergent growth method was the first one proposed and is currently the most widely used. This method arises from the seminal work of Tomalia and Newkome, as well as the branched model work of Vögtle [53]. In the divergent process, the construction of the dendrimer is starting from the core up to the periphery. This method requires two essential steps: (i) coupling of monomers, and (ii) activation of the monomer end-group, to promote the reaction with a new monomer [24]. The divergent growth method consists of the repetition of the two aforementioned steps, until the obtention of the desired dendrimer generation, as represented in Figure 4.

The divergent processing starts with activation or modification of the core and coupling of the first monomer, creating the first generation of the dendrimer. The next step is the deprotection or activation of this first generation (G1) to react with other branched monomers in order to couple the second generation (G2), and so on. When a new layer of branching units is created, a new generation is obtained, i.e., the number of the generation corresponds to the number of branched layers from the core [54]. In the divergent method, it is important that every step of the reaction is fully completed before the addition of a new generation so as to avoid deficiently formed branches [11]. The surface of the dendrimer may be easily functionalized and modified at each step, obtaining the desired pharmaceutical excipient at the end of the synthesis.

Usually, the divergent approach leads to the synthesis of highly symmetric dendrimer molecules. However, recently, researchers have taken up the possibility to create heterogeneously functionalized dendrimers by the divergent growth method, leading to dendrimers with several types of functional groups bound to the surface [24].

##### Convergent Growth Method

An alternative method used for the synthesis of dendrimers is the convergent process, proposed by Fréchet and Hawker in 1989–1990 [11]. In contrast to the divergent process, the convergent method synthetizes dendrimers starting from what will eventually become the exterior of the structure, i.e., the surface, and not from the core (Figure 5).

The convergent growth method also includes the repetition of the coupling and activation steps in order to obtain the desired dendritic structure. Primarily, the surface groups, generally two, are coupled to a monomer to give the dendritic segment (dendron generation zero). The second step consists of the activation of this fragment so that it can react with other monomers, thus creating the first generation dendron, that is, a dendritic wedge. This synthetic procedure can be repeated to give larger generation dendrons and use them to be coupled in the final step to a multifunctional core, producing the final dendrimer. The final part of the convergent synthesis ends up at the core, where two or more dendrons are joined together, creating the dendrimer. As the coupling reaction occurs at the focal point of the growing dendron, the preparation of large dendrimers (usually above the sixth generation) is difficulted by steric inhibition, resulting in decreased yields [52,53].

In convergent synthesis, greater structural control is achieved than with the divergent approach due to its relatively lower number of coupling reactions at each growth step, allowing the synthesis of dendritic products of unmatched purity. In addition, this strategy enables synthesis of asymmetric dendrimers, where different segments are coupled together to create dendrimers with heterogeneous morphologies, e.g., JDs. Due to the referred advantages, this synthesis process opens to intriguing fields of incorporating several active sites in one dendrimer to create a multifunctional excipient [24].

#### 2.2.2. Accelerated Approaches

There is a large number of reports about dendrimer synthesis. However, only a few have reached the market. This is justified by the large number of reaction steps that not only increase the chemical waste of valuable starting materials but also increase the probability of introducing structural defects in the dendritic form. Therefore, the dendrimer production becomes slow and costly. The most important difficulty when a high-generation dendrimer is synthesized is to ensure the full substitution of all reactive groups in order to avoid defects within the structure. To overcome those problems, accelerated approaches were developed with the intention to minimize the number of reaction steps, the reaction time, the starting materials, and the production associated costs [52].

##### Double Exponential Growth Technique

A fundamental breakthrough in the synthesis of the dendrimer is the double exponential growth technique, introduced by Moore in 1995. This method is a mixture of both the divergent and the convergent method. The double exponential technique allows formation of two types of monomers that are prepared by the convergent and divergent growth methods from a single starting material, as represented in Figure 6 [55,56].

Theoretically, the double exponential strategy requires a XY2 monomer with protected X and Y functional groups. Then, this monomer is activated selectively, i.e., the focal point or the periphery, to give two differently activated monomers, which are coupled together to obtain a G2 protected dendron. Repeating the selective activation and coupling process allows the formation of a G4, then a G8 dendron, and so on. On the final step, the focal points of the dendrons are activated and coupled to a multifunctional core obtaining a true dendrimer. The double exponential growth technique is similar to the convergent method, being very versatile because of the possibility of supramolecular preparation, classical, or asymmetrical dendrimers [52].

##### Double-Stage Convergent Method or Hypercore Approach

The hypercore approach is derived from the classical convergent technique. This double-stage method consists of three simple steps, as represented in Figure 7: (i) low-generation dendrons, with protected terminal groups, are coupled to a multifunctional core through their focal point; (ii) terminal groups of the obtained dendrimer (hypercore) are activated or deprotected and (iii) dendrons (different or the same) react with the hypercore, leading to the desired dendrimer [52].

The hypercore method enables the formation of dendrimers with chemically differentiated external and internal branches, due to the use of two different types of monomers in the synthesis of the dendrons and the hypercore. When compared to the convergent technique, the double-stage method uses a hypercore, which reduces the steric hindrance and helps obtaining higher generations and monodisperse dendrimers [57].

##### Hypermonomer Method, or the Branched Monomer Approach

In the branched monomer approach, the monomers, called hypermonomers, have a higher number of functional groups than conventional monomers. Using these XY4 hypermonomers (sometimes XY8), dendrimers with a high number of functional groups are obtained in a few steps. A great advantage of this method is the ability to obtain high-generation dendrimers in a smaller number of steps (Figure 8) [58].

#### 2.2.3. Advantages and Limitations of Synthetic Methods

Nowadays, it is important that the excipients conform to the regulations and expectations of the pharmaceutical market. Ideally, they are produced on a large scale, at a low cost and with high quality standards, so that these compounds can guarantee the performance of the drug and optimization of the therapeutic effect.

The choice of the synthesis method of the dendrimer depends on the intended dendrimer structure, the industrial goals, and large-scale production feasibility, which will impact on the way the branching is introduced. In this way, for the obtained dendrimer to have quality and exert its function as an excipient, the advantages and limitations of each synthesis method should be considered. Table 3 summarizes some important aspects that should be considered for each type of synthesis used on the production of dendrimers as pharmaceutical excipients.

### 2.3. Physicochemical Properties

Dendrimers are a relatively new class of compounds and they are characterized by their unique molecular architecture and dimensions. In comparison with other types of delivery systems, the advantages of using dendrimers include: (i) three-dimensional and globular architecture, (ii) controllable structure and size, (iii) lower molecular volume when compared with linear polymers of similar molecular weight, and (iv) a perfect opportunity for a wide variety of applications, including drug encapsulation [59,60]. Additionally, these nanosystems have important physicochemical properties, which render them good candidates for pharmaceutical excipients. To better understand the potential of dendrimers as pharmaceutical excipients, a discussion of the physicochemical properties is necessary.

#### 2.3.1. Nanoscale Size

Due to the fact that dendrimers have dimensions on a nanometric scale and because they have other protein-like properties, the dendritic polymers can be recognized as artificial proteins with biomimetic properties [61]. The size and shape of the dendrimers are close to the proteins found in the human body. Dendrimers’ size can be controlled through molecular engineering to closely resemble enzymes, antibodies, and globular proteins. PAMAM dendrimer generations 3, 4, and 5, with ammonia cores, may present the same dimensions of insulin (3 nm), cytochrome C (4 nm) and hemoglobin (5.5 nm), respectively [62].

Another important feature is that the dendrimers can cross bio-barriers like the blood–brain barrier (BBB) and membranes of tumor cells. The nanometric scale and uniformity of size enhance their ability to cross cell membranes, while reducing the risk of undesired clearance from the body through the liver or spleen [63].

Therefore, due to this similarity and to their high permeability in biological membranes, these polymeric nanocarriers are excellent excipients because they are capable of transporting the drug effectively through the organism.

#### 2.3.2. Higher Solubilization Potential

Generally, dendrimers are soluble and, for this reason, these nanosystems have been studied as solubilizers of drugs with low solubility [64]. Ionic interactions, hydrogen bonding, and hydrophobic interactions are the mechanisms by which dendrimers exert their solubility enhancement. Therefore, dendrimers are capable of improving the solubility, biodistribution, and efficacy of a number of therapeutic agents as well as being used as diagnostic and imaging molecules in animal models bearing brain tumors [65].

The presence of many terminal groups is responsible for the high solubility and reactivity of the dendrimers. For higher generation dendrimers, their solubility property depends mainly on the properties of their surface groups. Dendrimers with a surface with hydrophilic groups are soluble in polar solvents, whereas dendrimers with hydrophobic groups are soluble in nonpolar solvents. However, in addition to the functional groups, there are other intrinsic properties that can also influence the solubility of the dendrimer: (i) the nature of the repetition units, (ii) the generation number, and (iii) the core. It is noteworthy that the type of medium in which the dendrimer is found will also influence the solubility of this nanocarrier. Therefore, during exposure of the carrier system to a drug, the drug may or may not encapsulate into the dendrimer depending on the medium properties. If the drug molecules are poorly soluble in water, and the dendrimer provides a more hydrophobic environment, the drug will tend to encapsulate [16,66].

#### 2.3.3. Monodispersity

One of the advantageous properties in using dendrimers as pharmaceutical excipients is precisely the obtention of monodisperse nanoparticles with low levels of impurities. Dendrimers are highly branched molecules with a well-defined structure, exhibiting a high degree of monodispersibility. This is possibly due to their controlled synthesis and the purification processes these dendritic polymers are subjected to during their synthesis [67].

The great advantage of monodispersibility is that it is possible to predict their pharmacokinetic behavior in a biological organism. The pharmacokinetic properties are one of the most important aspects that need to be considered for the successful pharmaceutical application of a dendrimer. For example, the pharmaceutical industry uses the pharmacokinetic information to select the best drug that suits a specific pathology, the vehicle, and the route of administration, aiming to achieve the maximum therapeutic response and the lowest possible toxicity [68].

#### 2.3.4. Low Viscosity

One of the most important properties of dendritic macromolecules is low viscosity. Dendrimers in solution have a significantly lower viscosity than linear polymers. Although viscosity increases with the number of monomers, in the dendritic macromolecules, from a certain generation (usually from generation 4), the viscosity decreases. Thus, higher generation dendrimers have more functional groups, with lower viscosity than the low-generation dendrimers. This behavior differs from linear polymers because, on these structures, the intrinsic viscosity increases continuously with the molecular mass. The low viscosity is an advantageous property because the preparation of the dendrimer–drug complex becomes easier, and the immediate release of drugs is facilitated [64,69].

#### 2.3.5. Multivalent Surface

Multivalence is a property that dendrimers possess and is an important characteristic that leads them to be considered a good excipient. Multivalency, or polyvalency, refers to the number of reactive zones that a dendrimer presents on its surface (terminal groups) in order to interact with biological receptor sites, such as proteins, polymers, cells, and viruses. The surface modification may allow the design of dendrimers to mimic biological exo-receptors, substrates, cofactors, or inhibitors. Free surface groups can form complexes or conjugates with drugs or ligands by using cross-linking agents. In general, these interactions are reversible and occur in both inhibition and activation of biological processes. In contrast to weak monovalent receptor–ligand binding, multivalent interaction may amplify the signal transduction. Such augmentation is due to the enhanced affinity and cooperativity between the receptor and the ligand [65,70].

As the dendrimer generation increases, the number of functional groups present on the surface also increases, favoring the interaction of these with the biological targets. Svenson et al. [68] confirmed that drugs transported and delivered by dendrimers show a greater therapeutic response when compared to conventional drugs, due to the polyvalence that the dendrimers demonstrate (Figure 9).

#### 2.3.6. High Loading Capacity

The structure of the dendrimers can be used to load and store a wide range of inorganic or organic molecules by: (i) absorption on the surface by electrostatic interactions, (ii) conjugation with the surface groups through covalent bonding, or (iii) encapsulation of the drug into the cavities of the dendrimer, as shown in Figure 10 [71].

Although the number of molecules incorporated into a dendrimer depends on the architecture of this nanocarrier, the loading capacity increases with the number of functional groups on the dendrimer surface. The number of terminal groups available for drug interactions doubles with each increasing dendrimer generation.

The presence of a large number of ionizable groups on the surface of the dendrimers, such as the amine and carboxyl groups, provides an opportunity for electrostatic attachment of numerous ionizable drugs. An example of this is the electrostatic interaction between PAMAM dendrimers and the nonsteroidal anti-inflammatory drug ibuprofen. It has been proved that approximately 40 ibuprofen molecules interact with a G4 PAMAM dendrimer at pH 10.5, leading to a considerable enhancement of drug solubility [72].

The covalent attachment of the molecules to the surface groups of dendrimers, through hydrolyzable or biodegradable linkages, offers the opportunity for greater control over the drug release. Such an event may be justified by the fact that this type of interaction is stronger and more difficult to break when compared to other attachments. Yang and Lopina [73] have conjugated Penicillin V with both G2.5 and G3 PAMAM dendrimers through amide and ester bonds, respectively. They demonstrated that the amide linkage provided more stability than liposome-based drug-delivery systems, whereas the ester linkage of the drug to the dendrimer was demonstrated to increase the drug circulation time in the body via hydrolysis.

The technique of drug encapsulation within a dendrimer may be a purely physical entrapment, or it can involve interactions with specific structures within the nanocarrier [74]. For example, the existence of atoms of oxygen and nitrogen in the internal structure of the dendrimers allows interactions by hydrogen bonds with the drug. Generally, the empty internal cavities of the dendrimer are hydrophobic, allowing interactions with poorly soluble drugs [75]. Encapsulation is a general technique for low molecular weight molecules and for bioactive molecules which, if carried on the surface of the dendrimer, induce undesired immunogenicity [76]. Table 4 summarizes examples of incorporation of the drug into the dendrimer as well as the observed effects.

#### 2.3.7. Conformational Behavior

Dendrimers, like biological macromolecules, respond to the surrounding chemical environment showing altered conformational behavior. Consequently, as its function is intimately related to its structure, it is important to be aware of the type of effect the surrounding environment exerts on the dendrimer.

##### Dendrimers and the Effect of pH

Dendrimers have been reported to act as solubilizing agents to host both hydrophilic and hydrophobic drugs. However, the mechanism of solubilization largely depends on the protonated/deprotonated state of the dendrimer. Amino-terminated PPI and PAMAM dendrimers have basic terminal groups as well as a basic interior. For these types of dendrimers, with interiors containing tertiary amines, the low pH (pH < 4) region generally leads to extended conformations, based on highly ordered structure. At this pH, the interior is getting increasingly “hollow” as the generation number increases, as a result of repulsion between the positively charged amines both at the dendrimer surface and the tertiary amines in the interior. In addition, at neutral pH, back-folding occurs which may be a consequence of hydrogen bonding between the positively charged surface amines and the uncharged tertiary amines in the interior of the dendrimer. At higher pH (pH ≥ 10) the dendrimer contracts as the charge of the molecule becomes neutral, acquiring a more globular structure based on a compact network. At this point, the repulsive forces between the dendrimer arms and the terminal groups reach a minimum [84].

##### Dendrimers and the Effect of Salts

The high ionic strength, i.e., high concentration of salts, has a strong effect on charged dendrimers, such as PPI, and favors a contracted conformation of dendrimers, with a high degree of back-folding, somewhat similar to what is observed upon increasing pH. At low salt concentrations, the repulsive forces between the charged dendrimer segments result in an extended conformation in order to minimize charge repulsion on the structure [85].

##### Dendrimers and the Effect of the Solvent

The solvation power of any solvent to solvate the dendrimer is a very important criterion when investigating the conformation state of a dendrimer. Generally, dendrimers of all generations exhibit a larger extent of back-folding with decreasing solvent quality, i.e., decreasing solvation. Nevertheless, the low-generation dendrimers show the highest tendency towards back-folding, as a result of poor solvation, when compared to the higher generation dendrimers. Chai et al. [86] studied the solvent effect on PPI dendrimers. They concluded that a nonpolar solvent (benzene), poorly solvates the dendrimers favoring intramolecular interactions between the dendrimer segments and the back-folding. However, a weak acid solvent (chloroform) can act as a hydrogen donor for the interior amines in a basic dendrimer such as PPI, leading to an extended conformation of the dendrimer. This is due to the hydrogen bonding between the solvent and the dendrimer amines. Studies on polar dendrimers, such as amino-terminated PPI and PAMAM dendrimers, show the tendency that nonpolar solvents (“poor”) induce higher molecular densities in the core region as a result of back-folding, whereas polar solvents (“good”) solvate the dendrimer arms and induce a higher molecular density on the surface of the dendrimer.

## 3. Biodistribution and Toxicity

Nanosystems, especially dendrimers, have been used to overcome certain limitations of most conventional drugs such as (i) low water solubility, (ii) narrow therapeutic index, (iii) low concentration at the target, (iv) high affinity to plasma proteins, (v) rapid elimination of the drug, and (vi) low specificity on the biodistribution. For the dendrimer to be considered a good excipient, it needs to be able to overcome the biological barriers of the organism. The size, chemical composition, surface structure and the shape of the dendrimer influence both its biodistribution and its toxicity. In addition, these properties allow us to understand how they are metabolized and what is the long-term impact of the use of dendrimers at a cellular level [87].

### 3.1. Biodistribution of Dendrimers

Some in vivo studies have been performed to evaluate the biodistribution of dendrimers administered by the parenteral route. Kukowska-Latallo et al. [88] analyzed, in vivo, the biodistribution and elimination of G5 PAMAM dendrimer coupled to folic acid and the radioactive tritium marker (G5-3H-FA). In this study, two different dendrimers were synthesized and labelled with a radioactive compound: a dendrimer A, which was coupled to folic acid (G5-3H-FA), and a dendrimer B (control), which was not coupled to folic acid (G5 3H). They concluded that, on the first four days after administration, the clearance of G5 3H-FA dendrimers was lower than G5-3H dendrimers because the dendrimer A was found in tissues expressing the folic acid receptor. The kidney, being the main organ responsible for the elimination of these dendrimers, also expresses high amounts of the folic acid receptor. Thus, the kidney levels of the G5-3H dendrimers rapidly decreased, whereas the dendrimer A slightly increased on the first 24 h, due to the presence of the folic acid receptors on the renal tubules.

The biodistribution was evaluated on other studies, whose conclusions indicated that lower generation dendrimers (G3–G4) are cleared exclusively by the kidneys without further metabolization, G5 both by direct excretion by the kidney and excretion after liver conjugation, and higher generation dendrimers (G6–G9) are excreted only after hepatic metabolization. Therefore, it is possible to control the mode of excretion of a dendrimer by changing the number of generations of the nanoparticle, providing the dendrimer with an advantageous property as a pharmaceutical excipient [89].

Another study was performed with I^125^-labelled PAMAM dendrimers, in vivo, to study the biodistribution of this nanocarrier. Dendrimers with charged, either anionic or cationic surface groups, and hydrophobic dendrimers are rapidly cleared from the circulation, particularly by the liver [90]. However, Malik et al. [91] have further demonstrated that the anionic dendrimers remain in blood circulation for a longer period of time than the cationic dendrimers. In their turn, dendrimers with a hydrophilic surface (e.g., hydroxy-terminated or PEGylated dendrimers), and dendrimers with a higher number of generations, remain in circulation for longer periods.

Therefore, biodistribution is an important factor to be considered, and the dendritic conjugates should remain in the bloodstream for a sufficient period of time to achieve therapeutic efficacy and allow accumulation at the target, such as tumor cells. On the other hand, nanoparticles must be easily removed from the human body, in order to avoid unacceptable long-term accumulation [92].

### 3.2. Toxicity of Dendrimers

Although there are numerous advantages in the use of these nanosystems as a pharmaceutical excipient, it is important to evaluate the toxicity associated with dendrimers. Due to the size of the dendrimers (1–100 nm), they interact with some cellular elements, such as the cell membrane, nucleus, and proteins, as these cellular constituents are on the same dimension span. Furthermore, the dendrimers may also complex some metal ions, such as iron and zinc, affecting the biological action of hemoglobin and the renal function, respectively. However, the main determining factor for dendrimer-induced toxicity is its surface charge. The toxicity of a polymer in vivo is influenced by pharmacokinetics and biodistribution. In this way, biodistribution tests become indispensable to analyze which tissues or organs have a greater storage capacity of the drug and which, therefore, are potential targets of toxicity [87,90]. In the next sections, some important toxic effects will be mentioned in more detail.

#### 3.2.1. Membrane Interaction

Some investigators have demonstrated that cytotoxicity is highly related to the terminal groups present on the surface of the dendrimer. Lee and Larson [93] demonstrated that these cationic nanostructures interact with the negative charges of the phosphate groups of the lipid bilayer through electrostatic interactions, leading to the formation of small pores (nanopores), contributing to lower stability and increased cellular permeability (Figure 11). In addition, they concluded that cell membrane rupture is more pronounced at higher concentration, molecular weight, and generation of the dendrimer because these features are intimately related to the ability of the nanosystem to form nanopores. Thus, the results indicate that the spheroidal form of the dendrimers seems to be more efficient in drug transport than linear polymers, due to the increased permeability of the membrane. However, this increase in permeability can also lead to harmful effects and, ultimately, cellular lysis. It should be noted that other researchers also realized that the induction of permeability by dendrimers was not permanent, and that leaked cytoplasmic proteins returned to normal levels upon removal of dendrimers [94].

#### 3.2.2. Hemolytic Toxicity

The dendrimers which exhibit terminal cationic groups on their surface, interact with the red blood cells, leading to hemolysis. Bhadra et al. [95] performed an in vivo study of toxicity and concluded that the main limitation of cationic amine-terminated PPI dendrimers was the hemolytic effect. They concluded that dendrimers with free amine groups on their surface caused 35.7% and 49.2% of hemolysis for PPI G4 and G5 dendrimers, respectively. On the other hand, when the PPI dendrimers are peripherally coated with galactose, there is a significant reduction in the hemolysis to 10% and 7.1%, respectively, in the dendrimers G4 and G5. In contrast, studies with anionic dendrimers show an absence of hemolytic activity. Some authors suggested that the higher the generation of the cationic dendrimer, the greater the observed hemolysis, attributing this direct proportionality to the fact that these dendrimers have higher cationic charge [90].

#### 3.2.3. Cytokine Release

Dendrimers can modulate the release of the reactive oxygen species (ROS) and, therefore, increase cytokine production, which may be useful as a therapeutic tool or lead to significant toxic effects. Naha et al. [96] studied the inflammatory mediators, namely, the macrophage inflammatory protein-2 (MIP-2), tumor necrosis factor-α (TNF-α) and interleukin 6 (IL-6). These inflammatory mediators were measured by the enzyme linked immunosorbent assay (ELISA) after exposure of macrophage cells to PAMAM dendrimers (G4, G5, and G6). The investigators demonstrated that the toxic response of the PAMAM dendrimers correlated well with the number of surface primary amino groups, i.e., with the increase of the dendrimer generation. The response consists of an increased intracellular ROS production and cytokine-induced cytotoxicity which, in high concentrations, can lead to cell death.

It has been shown that both the anti-inflammatory properties and the toxic effect of dendrimers are intimately related to their terminal groups, charge, and number of generations. Avti et al. [97] demonstrated that PAMAM dendrimer terminated with amine (-NH2), hydroxyl (-OH) or carboxylic (-COOH) groups conjugated to glucosamine inhibit the release of cytokines. As a consequence, these dendrimers have potential as tools for various therapies, such as in rheumatoid arthritis. Although there is not enough information to establish the structure–activity relationships of dendrimers, these nanosystems show great potential in the future as possible anti-inflammatory therapeutic agents.

#### 3.2.4. Immunogenicity

Several studies aimed to test whether dendrimers exhibit an exaggerated immune response when administrated in the body. However, the literature suggests that dendrimers, depending on the size and end groups of the dendrimer´s surface, may provide a weak immunogenic response or even no response at all, not inducing the production of specific antibodies against the dendrimers. Roberts et al. [98] examined the immunogenicity of PAMAM dendrimers but observed no signs of immunogenicity within the dose range of 0.1–0.0001 µM. Agashe et al. [99] investigated in vivo the immunogenicity of G5 PPI dendrimers using ELISA for monitoring the antibody production. They reported that dendrimers were unable to provoke any detectable humoral immune response under the experimental conditions. This means that these nanosystems are not recognized by the host immune system as “foreign” particles, rendering them suitable for drug transportation across the body.

### 3.3. Solutions for Toxicity Issues

After the recognition that some dendrimers could induce cytotoxicity and hemolytic effect, the investigators developed several methods to overcome the adverse effects associated with the use of dendrimers [94]. Figure 12 provides a general overview of the various strategies that can be used to minimize the toxicity associated with dendrimers, which can be organized on biocompatible or biodegradable dendrimers and surface engineered dendrimers.

#### 3.3.1. Biocompatible or Biodegradable Dendrimers

One of the possible ways to decrease the toxicity associated with the dendrimers is based on the development of biodegradable central core, branches, and surface groups. However, researchers believe that the construction of dendrimers with biodegradable surface functionality is probably the most effective strategy to solve the toxicity issues of the dendrimer. These dendrimers are intended to be degraded into non-toxic compounds and subsequently eliminated from the circulation. Consequently, the use of these dendrimers allows the performance of their functions as pharmaceutical excipients without toxic or immunological effects [100]. Some examples of biocompatible dendrimers include: (i) the peptide dendrimer, ii) the polyester dendrimer, (iii) the triazine dendrimer, (iv) the phosphate dendrimer, v) the polyether dendrimer, (vi) the melamine dendrimer, and (vii) the polyether imine dendrimer [94]. These dendrimers have been studied and the investigators have been able to demonstrate that these nanocarriers can be used as drug-delivery vehicles in a safe and effective manner. These biocompatible dendrimers are mentioned in more detail in Table 5.

#### 3.3.2. Surface Engineered Dendrimers

Surface engineering appears to be one of the best strategies for decreasing dendrimers’ toxicity. This strategy is based on the modification of the surface groups in order to protect the cationic groups such as the amine groups. By using neutral or anionic molecules, the electrostatic interactions of these molecules with the cell membrane are prevented, thus avoiding the cytotoxicity of the cationic groups. In addition to the decreased toxicity, this technique also enables: (i) improvement of drug encapsulation efficiency, (ii) improvement of biodistribution and pharmacokinetic properties, (iii) an increase in solubility, (iv) targeting to specific site, (v) better transfection efficiency, (vi) sustained and controlled drug release, (vii) improvement of the stability profile, and (viii) improved potential of the anti-viral and anti-bacterial activity [94]. Table 6 presents several possible strategies for modifying the dendrimer surface groups.

## 4. Application of Dendrimers as Drug-Delivery Systems

Over the past years, growing attention has been drawn to the development of controlled and sustained drug-delivery systems. Dendrimers, due to their unique properties; like the globular shape, well-defined three-dimensional structure, high functionality, the presence of cavities, and small size; are suitable nanocarriers for drug-delivery applications. Consequently, these nanocarriers have stimulated wide interest in the field of nanotechnology, as the diverse biomedical applications, represented in Figure 13, testify.

The PAMAM dendrimer is the most well-studied and well-characterized class of dendrimers and was the first dendrimer that was synthesized and commercialized. For these reasons, this dendrimer has several medicinal and practical applications [27]. In the next sections, some applications of PAMAM and other dendrimers as drug-delivery systems will be mentioned.

### 4.1. Dendrimers in Ocular Drug Delivery

The topical application of APIs to the eye is the most prescribed route of administration for the treatment of various ocular disorders. However, the intraocular bioavailability of topically applied drugs is extremely poor. This is due to the drainage of the excess fluid via the nasolacrimal duct and the elimination of the solution by the tears. By using specialized delivery systems such as dendrimers, the referred difficulties of ocular drug delivery are diminished. Ideal ocular drug-delivery systems should be sterile, non-irritating, isotonic, biocompatible, biodegradable and should not run out from the eye. Trivedi et al. [116] improved the bioavailability of the pilocarpine in the eye recurring to dendrimers, and the results demonstrated that the resident time of the drug increased by using PAMAM dendrimers with hydroxyl or carboxylic groups.

### 4.2. Dendrimers in Oral Drug Delivery

Generally, the oral route is considered the favorite via for drug administration. Dendrimers are suitable candidates in oral drug delivery because these nanosystems improve drug solubility and absorption. Oral drug-delivery studies using the human colon adenocarcinoma, Caco-2 cell line, have shown that low-generation PAMAM dendrimers cross cell membranes. Additionally, P glycoprotein efflux transporter does not appear to affect dendrimers. Therefore, the drug dendrimer complex is able to bypass the efflux transporter [72]. In another study, Kolhe et al. [117] synthesized a fourth-generation PAMAM (PAMAM-G4-OH) dendrimer covalently linked to ibuprofen. The results suggested that the dendrimer–ibuprofen complex improved the drug efficacy by enhancing cellular delivery and that the complex produced a more rapid pharmacological response when compared to pure ibuprofen. Furthermore, these dendrimer–drug conjugates can potentially be modified by attaching ligands and antibodies for targeted drug delivery. The oral uptake of a polylysine dendrimer was studied by Florence et al. [118] with maximum reported levels at 6 h after administration of 15% in the small intestine, 5% in the small intestine, and 3% in the blood.

### 4.3. Dendrimers in Intravenous Drug Delivery

The intravenous route is the most direct method for delivering drugs into the body. However, the poor water solubility of many drugs (especially anti-cancer drugs) limits the application of the intravenous administration route. Additionally, intravenous administration of these drugs may result in several side effects, such as hemolysis and cytotoxicity. Numerous studies have been made to develop new formulations that are suitable for the intravenous route. Dendrimer–drug formulation had proved to be a useful nanocarrier of drugs with low solubility in different routes of administration because they can provide drugs with greater water-solubility, bioavailability, and biocompatibility. Consequently, there is a growing interest concerning the application of dendrimers as targeting carriers in cancer therapy [75]. Malik and collaborators [119] conjugated the PAMAM dendrimer with cisplatin, a potent anti-cancer drug with some toxicity and poor water solubility. This conjugate showed, in vivo, increased solubility, decreased systemic toxicity, and enhanced permeation as well as being able to selectively accumulate cisplatin in solid tumors. A polyester dendrimer was conjugated with doxorubicin and put in an injectable formulation. This formulation showed the potential to cure C-26 colon carcinomas in mice [120].

### 4.4. Dendrimers in Pulmonary Drug Delivery

Lungs represent an attractive alternative route and site of drug administration due to their large surface area, thin alveolar region, extensive vasculature, as well as avoidance of the first-pass metabolism. This advantage leads to increased systemic bioavailability of the drug and more effective therapeutic action. Due to their unique structure, many types of dendrimers have been designed, developed and studied for pulmonary delivery of various therapeutics. These nanosystems have been demonstrated to have good potential as inhalable drug-delivery alternatives for the treatment of pulmonary disorders [121]. Bai et al. [43] studied the PAMAM dendrimer as a carrier for pulmonary delivery of enoxaparin, a low-molecular-weight heparin, to treat vascular thromboembolism. They concluded that the positively charged PAMAM dendrimers are a suitable nanocarrier for pulmonary delivery of enoxaparin, without damage to the lungs. Additionally, these investigators showed that heparin encapsulated in pegylated dendrimers has a longer circulating half-time and increased pulmonary absorption.

### 4.5. Dendrimers in Central Nervous Systems (CNS) Drug Delivery

The brain is a challenging organ for drug delivery because of the BBB that is the best gatekeeper, protecting the CNS of the exogenous substances. Consequently, the drug delivery to the brain is challenging because many drugs have inadequate solubility, lipophilicity, and limited bioavailability, and the BBB can block 98% of drugs. Due to the ineffectiveness of conventional drug therapies, finding ways to deliver therapeutic drugs to the CNS safely and effectively is indispensable. Nanomedicine has shown great potential for the treatment of many CNS diseases with nanocarrier delivery systems such as dendrimers. These nanocarriers have demonstrated promising properties in CNS drug delivery with low toxicity and low immunogenicity as well as increased solubility, stability, and permeability of drugs. Furthermore, dendrimers have more efficient paracellular and transcellular transport across the BBB, which makes them ideal carriers for targeting water-insoluble drugs to the brain [122]. Katare et al. [123] investigated the efficiency of water-insoluble antipsychotic drug haloperidol via the intranasal route using the PAMAM dendrimer. They demonstrated that the aqueous solubility of haloperidol was increased with the dendrimer-based formulation and showed a significantly higher distribution of haloperidol in the brain and plasma compared to a control formulation of the drug. This study demonstrated the potential of dendrimers in improving the delivery of water-insoluble drugs to the brain.

### 4.6. Dendrimers in Transdermal Drug Delivery

Dendrimers were designed to be highly water-soluble and biocompatible. It was demonstrated that dendrimers are able to improve drug properties, such as plasma circulation time and permeation through the skin, thus delivering drugs efficiently on transdermal formulations. A PAMAM dendrimer complex with a nonsteroidal anti-inflammatory drug, such as ketoprofen or indomethacin, could improve the drug permeation through the skin as penetration enhancers. Chauhan and colleagues [124] investigated the enhanced bioavailability of the PAMAM dendrimer by using indomethacin as the model drug in transdermal drug application. They demonstrated that the PAMAM dendrimer is effective as a drug-delivery system because this nanosystem increased the flux of indomethacin across the skin in vitro as well as in vivo.

### 4.7. Dendrimers in Nasal Drug Delivery

Intranasal delivery is distinguished from the various strategies currently available for drug targeting. It is non-invasive and reduces the exposure of non-target sites to the API, thus increasing the efficiency and safety of the drug. Drug molecules can be targeted to the brain via the nasal cavity through the trigeminal nerve pathway and the olfactory nerve pathway. Dendrimers have been reported to enhance the aqueous solubility of drugs by forming a complex with them. This complex would provide a high concentration of the drug in the nasal area. PAMAM dendrimers, for example, have caught the attention of researchers regarding nose-to-brain targeting [122]. Perez et al. [125] studied intranasal delivery with dendrimers by coupling radioactive small interfering ribonucleic acid (siRNA) to PAMAM dendrimers to form dendriplexes (siRNA−dendrimer complexes) and formulated these particles into mucoadhesive gels. Several concentrations of the different gels were tested, and no toxicity was observed. Moreover, dendriplexes showed increased radioactivity in the brain. Thus, this study demonstrated the potential of PAMAM dendrimers in improving the delivery of drugs to the brain via intranasal administration.

### 4.8. Dendrimers in Gene Delivery

The ability to transfer genetic material efficiently, into the cytoplasm and the nucleus of eukaryotic cells may allow the treatment of a variety of genetic disorders. Dendrimers are one of the most useful gene-delivery systems and play a significant role in the development of vectors for gene delivery due to their ability to transfect genes without inducing toxicity. Additionally, the high charge density in the surface of the nanocarrier allows optimal condensation and formation of nanostructures with deoxyribonucleic acid (DNA). Among the several commercially available dendrimers, PAMAM dendrimers have received the most attention as potential gene delivery agents due to their cationic nature which enables DNA binding at physiological pH by electrostatic interactions. Literature suggests that functionalized dendrimers are much less toxic than the native dendrimers [122]. Luo et al. [126] revealed the low cytotoxicity of PEG-modified PAMAM dendrimers and their efficiency on the DNA delivery to the cells. They demonstrated that the PEG-modified PAMAM dendrimer is an extremely efficient, highly biocompatible, low-cost DNA delivery system, and it can be readily used in basic research laboratories as well as in future clinical applications. On another study [127] water-soluble carbosilane dendrimers were complexed with oligonucleotides via electrostatic interaction. Perrisé-Barrios et al. [40] reported the synthesis of carbosilane dendrimers 2G-NN16 and 2G-03NN24 with the capacity to transfect siRNA to CD4 T lymphocytes, inhibiting HIV-1 replication.

### 4.9. Dendrimers in Vaccines

Most low-molecular-weight substances are not immunogenic and, consequently, when it is desired to raise induce antibody production against small molecules, they must be conjugated to a macromolecule. Nowadays, a possible alternative strategy to solve this problem is to use dendrimers as nanocarriers of these small antigens. Dendrimers have optimal characteristics as efficient immunostimulant compounds (adjuvants) that can increase the efficiency of vaccines. Several studies have been performed to verify if the PAMAM dendrimer is an ideal carrier of small antigens. The results demonstrated that this nanosystem does not induce adverse host responses, including immune and/or inflammatory reactions after administration. This demonstrates that PAMAM dendrimers can be used successfully in conjugates with antigens [122,128].

## 5. Patents

Although dendritic polymers have a history of nearly three decades, the number of papers and patents is increasing every year, which is indicative of the continuous progress of their applications in academic research, and industrial processes, as well as in the biomedical field [129]. In Table 7, some recent examples of patents for dendrimers as drug-delivery systems are mentioned.

## 6. Conclusions and Future Perspectives

Dendrimers, as excipients, are predicted to have a foremost role in both the pharmaceutical industries and for medicine. Due to their structural properties (such as nanoscale uniform size, the high degree of branching, polyvalency, water solubility, and availability of internal cavities), and the fact that they can be almost precisely controllable during their synthesis, dendrimers may be used as excipients to improve drug-formulation properties. Additionally, there is a possibility of extending the patent lifetime.

The drug may be attached to the dendrimer by covalent bonds, electrostatic interactions or encapsulation. The choice of drug–dendrimer interaction is dependent on the needs and properties of the drug and the type of pathology. Dendrimers are excellent drug carriers and can be carefully engineered for the delivery of biomolecules to target cells, allowing the use of smaller drug doses, with smaller side effects. 

Another advantage of the dendrimer is that this nanosystem has been shown in various studies, to be a compatible, safe, and effective nanocarrier in various routes of administration. Consequently, it allows us to increase the range of drugs that have therapeutic value but are rejected by the pharmaceutical industry because of their low water-solubility.

Many studies already did show evidence that dendrimers are promising excipients in several therapeutic formulations. However, it should be noted that the studies carried out so far have been performed on in vitro experimental models or animal models. Thus, it is not yet possible to extrapolate these results to humans. Although this new technology is promising, further in vivo studies are required to understand with more accuracy the biocompatibility and dendrimer-associated toxicity. Only then will clinical studies in humans be possible to evaluate the medium/long-term toxicological impact.

Although dendrimers are relatively new structures (approximately 30 years) and the fact is that they cannot be recognized as a pharmaceutical excipient yet, they present a promising future in the pharmaceutical and biomedical field.

## Figures and Tables

**Figure 1 materials-13-00065-f001:**
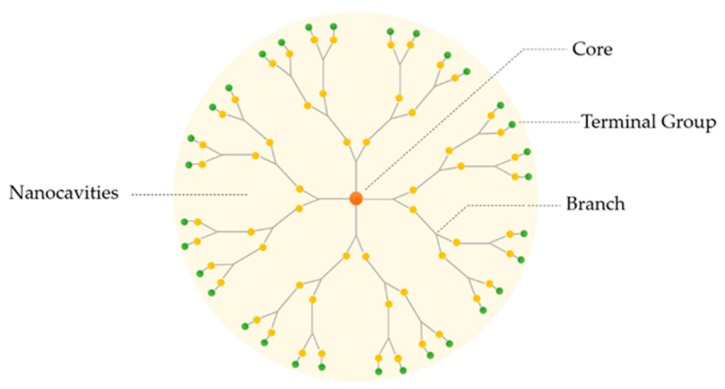
Basic structure of a dendrimer.

**Figure 2 materials-13-00065-f002:**
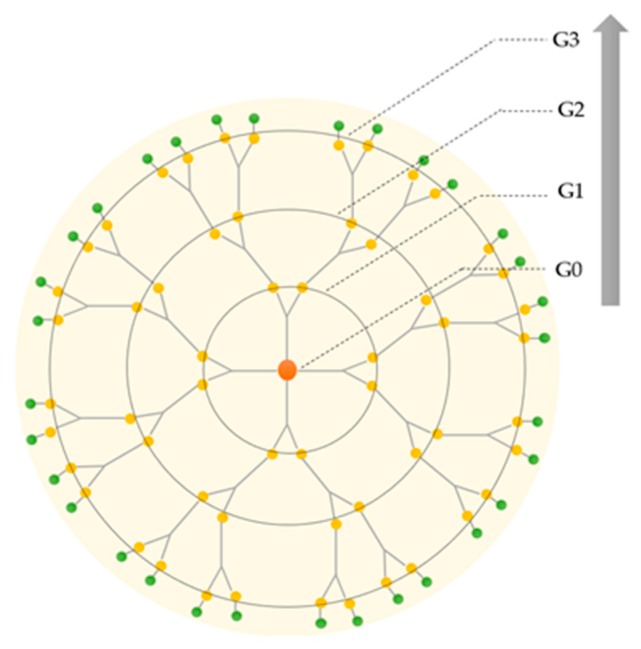
Schematic representation of the increasing generations of the dendrimer: from first (G0) to third generation (G3).

**Figure 3 materials-13-00065-f003:**
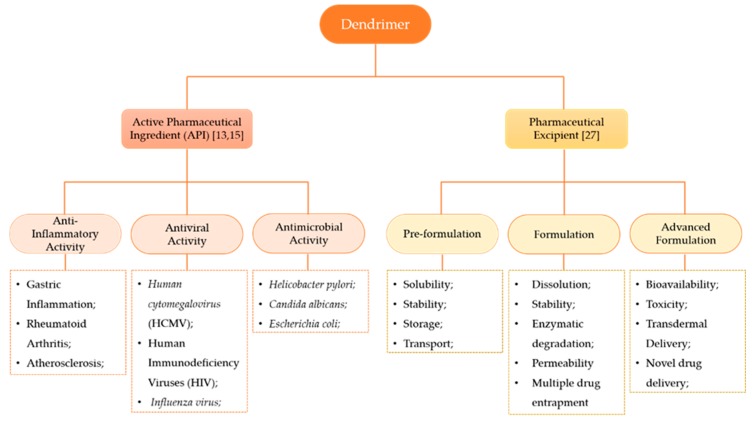
Dendrimers’ classifications according to their roles in the formulation. The constitution of the dendrimer may confer different activities, namely, anti-inflammatory activity, antiviral activity and antimicrobial activity [13,15]. As a pharmaceutical excipient, dendrimers may enhance distinct properties of the formulation according to the phase of drug product development [27].

**Figure 4 materials-13-00065-f004:**
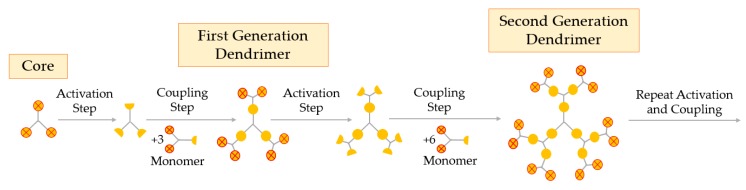
Synthesis of dendrimers by the divergent growth method.

**Figure 5 materials-13-00065-f005:**
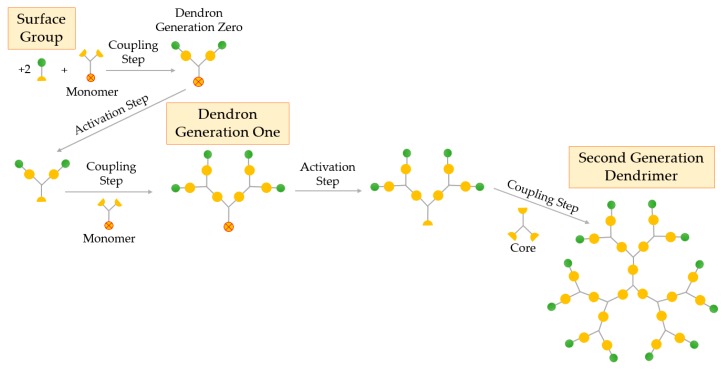
Synthesis of dendrimers by the convergent growth method.

**Figure 6 materials-13-00065-f006:**
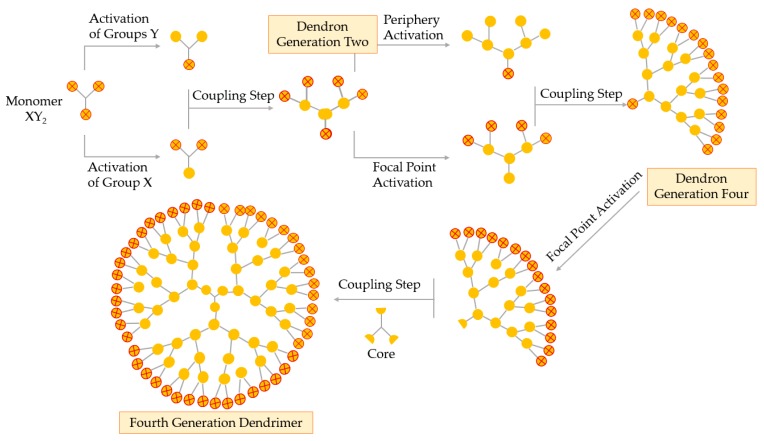
Synthesis of dendrimers by the double exponential growth technique.

**Figure 7 materials-13-00065-f007:**
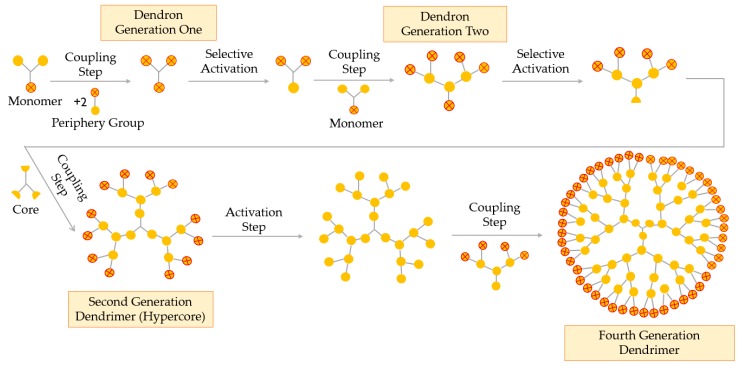
Synthesis of dendrimers by the double-stage convergent method or the hypercore approach.

**Figure 8 materials-13-00065-f008:**
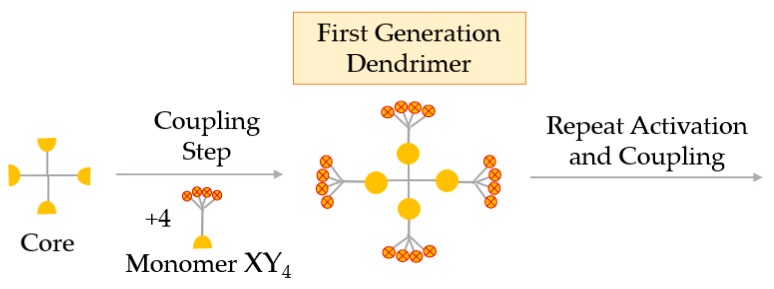
Synthesis of dendrimers by the hypermonomer method, or the branched monomer approach.

**Figure 9 materials-13-00065-f009:**
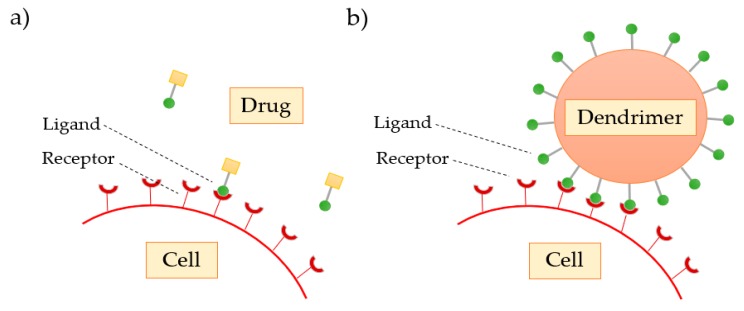
Comparison of the (**a**) classical interaction of the free drug with the cell receptor with the (**b**) enhanced dendrimer interaction.

**Figure 10 materials-13-00065-f010:**
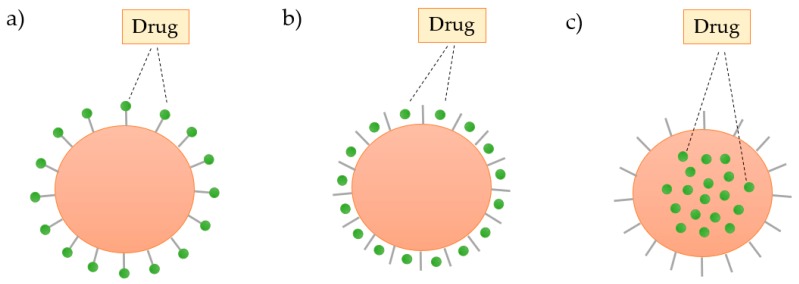
Schematic representation of the three ways of incorporation of the drug in the dendrimer: (**a**) covalent binding, (**b**) electrostatic interactions, and (**c**) encapsulation.

**Figure 11 materials-13-00065-f011:**
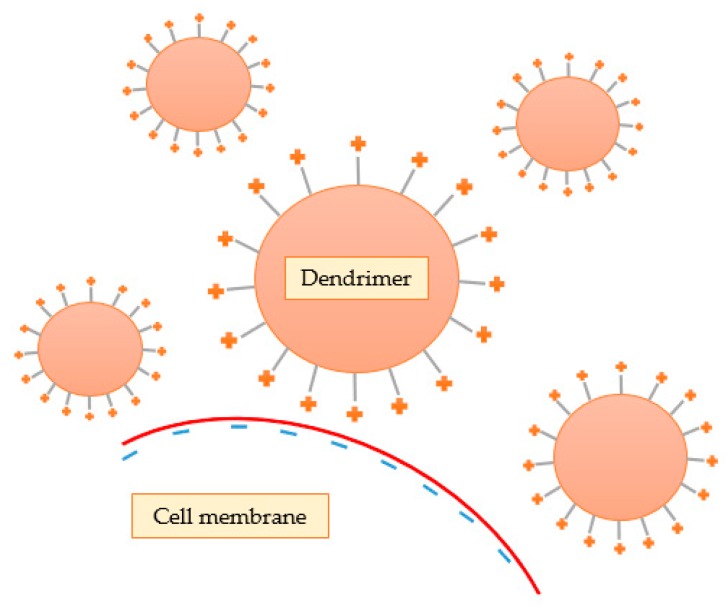
Cationic dendrimers interact with the negative charges of the lipid bilayer through electrostatic interactions, leading to the formation of nanopores.

**Figure 12 materials-13-00065-f012:**
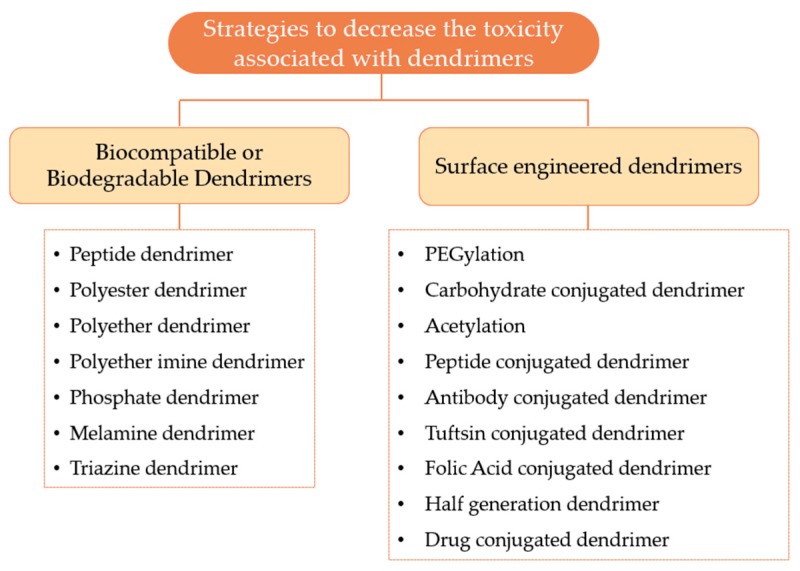
Various strategies to decrease the toxicity related to dendrimers.

**Figure 13 materials-13-00065-f013:**
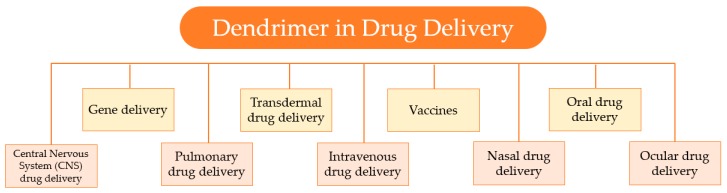
Several applications of dendrimers in drug delivery.

**Table 1 materials-13-00065-t001:** Summary of different types of dendrimer with potential for being excipients, their chemical structure and their use in pharmaceutical formulations.

Dendrimer Name	Chemical Structure	Mechanism	Reference
Poly(propyleneimine) (PPI) dendrimers	Terminal groups with primary amines and the interior of PPI contains tertiary tris propylene amines.	Increased drug solubility through electrostatic interactions.	[36]
Frechet-type dendrimers	Hyper-branched architecture of polybenzyl ether. Contains -COOH groups as terminal groups.	Helps to enhance solubility in aqueous media and other polar solvents.	[25]
Peptide dendrimers	Peptidyl branching core and/or covalently attached as surface functional units.	Acts as surfactant and carrier for drug and gene delivery.	[25,37]
Glycodendrimers	Contains saccharide residues as terminal groups and a core with sugar units.	Site-specific drug delivery to the lectin-rich organs.	[25,38]
Hybrid dendrimers	A blend of linear and dendritic polymers.	Acts as a surfactant and drug-delivery system.	[25]
Polyester dendrimers	Polyester-based dendrimers.	Drug targeting, improved biodistribution, and modulation of drug release.	[25,39]
Poly-L-lysine (PLL) dendrimers	Core and branching units are based on the amino acid lysine.	Gene carriers and increased drug solubility.	[11]
Carbosilane dendrimers	Si-based dendrimers, allowing functionalization and stability.	Gene therapy enhancer	[40]

**Table 2 materials-13-00065-t002:** Overview of the effects of the use of poly(amidoamine) (PAMAM) dendrimers in different pharmaceutical applications.

API	Application	Observed Effects	Reference
Pilocarpine	Ocular drug delivery	Improved residence time of pilocarpine in the eye	[42]
Enoxaparin	Pulmonary drug delivery	Increased bioavailability of enoxaparin by 40%	[43]
Ketoprofen	Transdermal drug delivery	Improved drug permeation through the skin	[44]
Diflunisal	Transdermal drug delivery	Improved drug permeation through the skin	[44]

**Table 3 materials-13-00065-t003:** Summary of the main advantages and disadvantages of the dendrimer preparation method.

Method of Preparation	Advantages	Disadvantages	Examples of Dendrimers
Divergent growth method	Fast synthesis; production of large quantities; synthesis of highly symmetric dendrimers; the surface of the dendrimer can be easily modified with desired functional groups; allows the formation of high-generation dendrimers.	Possibility of defects in the higher generation dendrimers product; difficult in the separation of the desired product from reactants; excess of reagents; requires numerous steps to form a large structure; requires a large quantity of starting material; possible incomplete reaction of the terminal groups.	PAMAM; PPI; Poly(arylalkyl ether);
Convergent growth method	Easy to purify the desired product; the occurrence of defect is minimized; possibility of synthesis of asymmetric dendrimers; involves only a small number of reactions per molecule; provides greater structural control than the divergent approach.	Does not allow the formation of a high generation of dendrimers; lower yield; difficult to modify the terminal groups.	JDs; Poly(aryl ether); Poly(aryl alkyne); Poly(phenylene); Poly(alkyl ester); Poly(alkyl ether);
Double exponential growth technique	Elaboration of large multifunctional dendrons or dendrimers; preparation of symmetric, supramolecular, or asymmetrical dendrimers; high synthetic yields; a large number of dendrimers using the same monomers for 2–3 times.	The process is time-consuming, as the method uses both convergent as well as divergent processes.	Poly(phenylacetylene); Poly(amide); Poly(ether urethane); Poly(ester); JD;
Double-stage convergent method	Allows the formation of high-generation dendrimers; uses a hypercore that reduces the steric effect; helps to obtain more monodisperse dendrimers; enables the formation of dendrimers with chemically differentiated internal and external branches.	The synthesis of the hypercore, the dendrons, and the final dendrimers is slow.	Phenylacetylene; Poly(amide);
Hypermonomer method	Dendrimers showing a high number of functional groups in fewer steps; allows the formation of high-generation dendrimers in a few steps.	Synthesis requires several growth and activation steps; the acceleration is limited to generating dendrimers; monomer synthesis is a time-consuming process.	Poly(aryl ether); Triazine;

**Table 4 materials-13-00065-t004:** Examples of incorporation of the drug into the dendrimer as well as the observed effects.

Dendrimer	Drug Loaded	Formulation Type	Results	Reference
PEG-PAMAM-G4	Silybin	Encapsulation	Increased solubility.	[77]
PAMAM-Biotin	SB-T-1214	Conjugation	High potency and targeted drug delivery.	[78]
PAMAM-G4-DHA	Paclitaxel	Conjugation	Increased pharmacological activity in upper gastrointestinal cancer.	[79]
PAMAM	Berberine	Conjugation and Encapsulation	Improved pharmacokinetic profile.	[80]
PAMAM	Gallic acid	Conjugation	Improved bioavailability.	[81]
Silica-PAMAM	Black Carrot Anthocyanin	Encapsulation	Sustained release; less toxicity and enhanced activity.	[82]
PAMAM-G4	Resveratrol	Encapsulation	Improved solubility.	[83]

**Table 5 materials-13-00065-t005:** Examples of biocompatible dendrimers as well as the observed effects.

Biocompatible Dendrimer	Chemical Structure	Results	Reference
Peptide dendrimer	PLL-Lactose G4 dendrimer	Reduces hemolysis.	[101]
Polyester dendrimer	Polyester dendrimer with ethylene oxide as the branching unit	Absence of toxicity in cells and decreased drug toxicity.	[102]
Polyether dendrimer	Carboxylate and malonate as terminal groups	Absence of hemolysis in the erythrocytes one hour after its administration.	[91]
Polyether imine dendrimer	Carboxylic acid as a terminal group	Absence of toxicity in cells.	[103]
Phosphate dendrimer	5G thiophosphate dendrimer	The dendrimer is neither hemotoxic nor cytotoxic.	[104]
Melamine dendrimer	Melamine as the branching unit	Significant reduction in hepatotoxicity.	[105]
Triazine dendrimer	Triazine dendrimer with hydrazone linkages	No toxic effect and degradable into small molecules.	[106]

**Table 6 materials-13-00065-t006:** Examples of strategies to modifying the dendrimer surface groups.

Technique	Conjugated Molecule	Results	Reference
PEGylation	Polyethylene glycol (PEG)	Improved drug loading and decreased hemolytic toxicity of the PAMAM dendrimer.	[107]
Carbohydrate-conjugated dendrimer	Maltose	Decreased hemolytic activity inherent to the PPI dendrimers.	[108]
Acetylation	Acetyl groups	Decreased PAMAM dendrimers’ toxicity and maximized their transepithelial permeability.	[109]
Half generation	Carboxylic groups	Decreased cytotoxicity associated with the PAMAM dendrimer.	[110]
Peptide-conjugated dendrimer	Arginine-glycine-aspartate peptide	The conjugation of tripeptides minimized the cytotoxicity of the cationic PAMAM dendrimer.	[111]
Drug-conjugated dendrimer	Flurbiprofen	The drug-dendrimer complex showed lesser hemolytic toxicity than the PAMAM dendrimer.	[112]
Antibody-conjugated dendrimer	Human epidermal growth factor receptor-2 monoclonal antibody (Anti-HER2 mAb)	Rapid and efficient cellular internalization of the dendrimer-antibody conjugated with low systemic toxicity.	[113]
Tuftsin-conjugated dendrimer	Threonyl-lysyl-prolyl-arginine peptide (Tuftsin)	Tuftsin–PPI complex possessed lower cytotoxicity than the PPI dendrimer.	[114]
Folic acid-conjugated dendrimer	Folic acid and Polyethylene glycol (PEG)	Folic acid–PEG-PAMAM has lower hemolytic toxicity compared to the PEG-PAMAM and the PAMAM dendrimer.	[115]

**Table 7 materials-13-00065-t007:** Examples of patents of dendrimers as drug-delivery systems.

Pharmaceutical Application	Dendrimer	Drug loaded	Summary	Publication Date	Patent	Reference
Gene delivery	PAMAM	MicroRNA-150 (miR-150)	A PAMAM dendrimer was designed for sustained delivery of miR-150 to FLT3-overexpressing acute myeloid leukemia cells. Preclinical animal model studies have demonstrated good therapeutic efficacy.	2019	US20190175754	[130]
CNS drug delivery	PAMAM	Prion protein (PrP)	PrP was conjugated to PAMAM dendrimers for Alzheimer’s therapy. This complex will inhibit β-amyloid plaque formation (they act as potent neurotoxins in vitro and in vivo in Alzheimer’s disease).	2019	US20190092837	[131]
Tumor drug delivery	PAMAM	Disulfiram	Disulfiram and photosensitizer indocyanine green were entrapped into PAMAM-G0 dendrimer for anti-tumor therapy. This prepared a nanodrug-delivery system that can simultaneously play roles of chemotherapy and photodynamic therapy.	2018	CN108888764	[132]
Tumor targeting and controlled drug release	PAMAM	Doxorubicin(DOX)	Tumor targeting and controlled drug release of the DOX-PEG-PAMAM dendritic complex is controlled by the pH.	2017	CN107596385	[133]
Tumor targeting	PAMAM	Erlotinib	The Erlotinib–PAMAM dendrimer will target tumor cells with a high expression of CD44 and can specifically deliver more drugs to the tumor site.	2017	CN107281164	[134]
Targeted drug delivery	PLL	Polynucleotides	The rabies virus glycoprotein (RVG) was conjugated to the PLL dendrimer to provide effective and safe delivery of polynucleotides to target cells.	2012	KR1020120067168	[135]
Gene delivery	PLL	Plasmid DNA	A PLL system containing a vector with intracellular nuclear protein binding and reducible polymers is provided to stabilize plasmid DNA in an extracellular region, and to promote its absorption to the target cell.	2012	KR1020120007208	[136]
Cancer targeting	Peptide-dendrimer	Docetaxel	The peptide was conjugated with the dendrimer for targeting, imaging, and treatment of prostate cancer.	2018	EP3402484	[137]
Vaccine	Positively charged dendrimer	Antigen	Branched polymeric dendrimers (e.g., PAMAM and other dendrimers) were used as vehicles for the targeted delivery of antigen to specific cells, giving rise to a new nanoparticle-based method for genetic or protein vaccination.	2018	US20180099032	[138]
Drug-delivery system	Asymmetric dendrimer	Paclitaxel	Paclitaxel-loading asymmetric dendrimer nanometer drug carrier system has the anti-tumor treatment index and biosecurity enhanced compared with those of free Paclitaxel during the in vivo treatment.	2017	CN106512021	[139]
Transdermal drug delivery and permeation enhancer	Second-generation oleodendrons	Diclofenac	Oleic acid-based dendron is used as a potential chemical penetration enhancer in transdermal drugs.	2013	IN1749/MUM/2010	[140]

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
