# Peer review of "Dendrimers as Pharmaceutical Excipients: Synthesis, Properties, Toxicity and Biomedical Applications"

_materials, 2019, doi:10.3390/ma13010065_

Round 1

Reviewer 1 Report

The manuscript reviews the application of dendrimers for therapy, as excipients that help to overcome  limitations of certain drugs or for gene delivery. The subject is of high interest and deserves a review, indeed. However, for this manuscript to be accepted I would ask the revision of some statements that are not clear or exact:

105-108: Not all Janus dendrimers are necessarily amphiphilic. There is a review by A.M. Caminade et al. that should be included (New J. Chem., 2012,36, 217-226.)

127-128: What is the meaning of this definition: “Dendrimers or dendritic polymers are nanoparticles based drug deliverysystems, with a size between 1 nm to 100 nm”? All dendrimers and dendritic polymers are drug delivery systems? All dendrimers or dendritic polymers are nanoparticles? It is a confusing sentence.

144: “However, some of the dendrimers, such as PAMAMs, do not have proper spherical structure [31].” What do authors mean with this statement. Later, dendrimers are presented as globular objects. What is the difference. Reference 31 is a review on dendrimers and I havn’t found support that explains the meaning of this sentence.

195: The classification of dendrimers between PAMAM dendrimers and Janus dendrimers is rather confusing. PAMAM dendrimers are a group of dendrimers with the same polyamidoamine chemical structure. In contrast, Janus dendrimers can be formed by dendrons with different chemical structures including PAMAM, Frèchet-type, etc. So, it is not easily deduced the rationale for this classification and also the classification presented in Table 2. Please revise this.

203-204: “There are two basic types of polymers that consist entirely of branched repeat units, namely, dendrimers and classical polymers.  The classical polymers exhibit an irregular architecture with incompletely reacted branch points throughout the structure.” What do authors refer to with the term “classical polymers”? Are they branched polymers? But, there many types of branched polymers: Side-chain polymers, star-like polymers, hyperbranched polymers, etc.  Why are they considered classical polymers? I would prefer the authors to use those terms/names accepted for the classification of polymers, depending on their architecture so that it is clear what type of polymers are being considered as a basic type of polymer to be compare with dendrimers.

650- 651: “In the next sections, some applications of PAMAM dendrimer as drug delivery systems will be mentioned. “:  The discussion about the different applications, and the issues to be taken into account with respect to their use as excipients for therapy, is mainly focused on PAMAM dendrimers. Isn’t it there worth–mentioning works about other dendrimers widely explored for this purpose: polyester dendrimers (bis-MPA derivatives), carbosilane dendrimers, etc. Moreover, the latter should me included in table 2.

Author Response

Ref: materials-656664

Title: Dendrimers as Pharmaceutical Excipients: Synthesis, Properties, Toxicity and Biomedical Applications

Dear Editor,

The authors appreciated the careful review and greatly acknowledge the comments that the Reviewers have provided on our manuscript. We have carefully revised the manuscript and have made the recommended changes and answered in detail to the questions raised. All changes made to the text are highlighted in yellow and grey colours. Please find below a point-by-point list of answers to the reviewers’ concerns.

Reviewer #1

105-108: Not all Janus dendrimers are necessarily amphiphilic. There is a review by A.M. Caminade et al. that should be included (New J. Chem., 2012,36, 217-226.)

Response:

Thank you for the comment. The information was corrected accordingly. Please see line 167 and following.

127-128: What is the meaning of this definition: “Dendrimers or dendritic polymers are nanoparticles based drug delivery systems, with a size between 1 nm to 100 nm”? All dendrimers and dendritic polymers are drug delivery systems? All dendrimers or dendritic polymers are nanoparticles? It is a confusing sentence.

Response:

Thank you for calling our attention on this issue. The definition was clarified. Please, see line 100 and following.

144: “However, some of the dendrimers, such as PAMAMs, do not have proper spherical structure [31].” What do authors mean with this statement? Later, dendrimers are presented as globular objects. What is the difference? Reference 31 is a review on dendrimers and I haven’t found support that explains the meaning of this sentence.

Response:

Thank you for the comment. The section was revised accordingly. Please, see line 116 and following.

195: The classification of dendrimers between PAMAM dendrimers and Janus dendrimers is rather confusing. PAMAM dendrimers are a group of dendrimers with the same polyamidoamine chemical structure. In contrast, Janus dendrimers can be formed by dendrons with different chemical structures including PAMAM, Frèchet-type, etc. So, it is not easily deduced the rationale for this classification and also the classification presented in Table 2. Please revise this.

Response:

Thank you for the comment. The classification was clarified. Please see line 171 and following

203-204: “There are two basic types of polymers that consist entirely of branched repeat units, namely, dendrimers and classical polymers.  The classical polymers exhibit an irregular architecture with incompletely reacted branch points throughout the structure.” What do authors refer to with the term “classical polymers”? Are they branched polymers? But, there many types of branched polymers: Side-chain polymers, star-like polymers, hyperbranched polymers, etc.  Why are they considered classical polymers? I would prefer the authors to use those terms/names accepted for the classification of polymers, depending on their architecture so that it is clear what type of polymers are being considered as a basic type of polymer to be compare with dendrimers.

Response:

Thank you for the comment. The terminology was revised. Please see section 2.2

650- 651: “In the next sections, some applications of PAMAM dendrimer as drug delivery systems will be mentioned. “:  The discussion about the different applications, and the issues to be taken into account with respect to their use as excipients for therapy, is mainly focused on PAMAM dendrimers. Isn’t it there worth–mentioning works about other dendrimers widely explored for this purpose: polyester dendrimers (bis-MPA derivatives), carbosilane dendrimers, etc. Moreover, the latter should be included in table 2.

Response:

According to the Reviewer’s comment Table 1 (former Table 2) was expanded and further examples were added to section 4.

Reviewer 2 Report

Authors have described the role  dendrimers as excipients in a very elaborated manner.

Due to their size and structure, they act flexible or manageable so qualify as an excellent candidate to be used as excipient in pharma and medicine industries.

Dendrimers as excipients in formulation have shown great compatibility there are safe to use.

Though this reviewer does not have any major comment and is very satisfied with the writing and information elaboration style, authors are suggested to read-proof the article once again for any grammar and typo errors.

Author Response

Ref: materials-656664

Title: Dendrimers as Pharmaceutical Excipients: Synthesis, Properties, Toxicity and Biomedical Applications

Dear Editor,

The authors appreciated the careful review and greatly acknowledge the comments that the Reviewers have provided on our manuscript. We have carefully revised the manuscript and have made the recommended changes and answered in detail to the questions raised. All changes made to the text are highlighted in yellow and grey colours. Please find below a point-by-point list of answers to the reviewers’ concerns.

Reviewer #2

Authors have described the role dendrimers as excipients in a very elaborated manner.

Due to their size and structure, they act flexible or manageable so qualify as an excellent candidate to be used as excipient in pharma and medicine industries.

Dendrimers as excipients in formulation have shown great compatibility there are safe to use.

Though this reviewer does not have any major comment and is very satisfied with the writing and information elaboration style, authors are suggested to read-proof the article once again for any grammar and typo errors.

Response:

Thank you for the comments. The article was revised. Please see the text highlighted in yellow colour.

Reviewer 3 Report

This is a nice review.

Everything is there, but not always in the best order. Some figures are not required. Here is my list of corrections and comments:

1) Page 1 line 34. replace 'diameter' with 'at least one of their dimensions'

2) Page 3. The entire section (lines 95-121) on PAMAM feels like it should come later. I think this would work better if section 2 'Dendrimers' was moved earlier to line 95.

3) Page 5. Figure 3. Would be useful to include a reference for each use on the diagram (as a numbered reference).

4) Page 17. Figure 12 is unnecessary.

5) Page 18.Figure 13 is unnecessary.

Author Response

Ref: materials-656664

Title: Dendrimers as Pharmaceutical Excipients: Synthesis, Properties, Toxicity and Biomedical Applications

Dear Editor,

The authors appreciated the careful review and greatly acknowledge the comments that the Reviewers have provided on our manuscript. We have carefully revised the manuscript and have made the recommended changes and answered in detail to the questions raised. All changes made to the text are highlighted in yellow and grey colours. Please find below a point-by-point list of answers to the reviewers’ concerns.

Reviewer #3

This is a nice review.

Everything is there, but not always in the best order. Some figures are not required. Here is my list of corrections and comments:

1) Page 1 line 34. replace 'diameter' with 'at least one of their dimensions'

Response:

Thank you for the comment. The information was corrected accordingly. Please see line 34.

2) Page 3. The entire section (lines 95-121) on PAMAM feels like it should come later. I think this would work better if section 2 'Dendrimers' was moved earlier to line 95.

Response:

The text was rearranged accordingly. Please see lines 175 and following (in grey).

3) Page 5. Figure 3. Would be useful to include a reference for each use on the diagram (as a numbered reference).

Response:

References were added accordingly. Please, see figure 3.

4) Page 17. Figure 12 is unnecessary.

Response:

Thank you for the comment. The figure 12 was removed.

5) Page 18.Figure 13 is unnecessary.

Response:

Thank you for the comment. The figure 13 was removed.

Round 2

Reviewer 1 Report

All my comments have been addressed but the revision must be done more carefully. Bibliography must be revised. For references 40, 116 and 119 the same paper has been cited. This paper does not correspond to what references 116 (page 683) and 119 (page 700) should deal with. The appropriate papers should be cited.

590-591: Figure 13 caption should be eliminated if figures 12 and 13 are not being included in the revised version. Figures 14 and 15 should be renumbered accordingly.

1111: ref 125 does not display the correct authors names and order.

In general, authors should revise English writing and correct some grammar and spelling mistakes

Author Response

Ref: materials-656664

Title: Dendrimers as Pharmaceutical Excipients: Synthesis, Properties, Toxicity and Biomedical Applications

Dear Editor,

The authors appreciated the careful review and greatly acknowledge the comments that the Reviewer#1 has provided on our manuscript. We have carefully revised the manuscript and have made the recommended changes and answered in detail to the questions raised. All changes made to the text are highlighted in green colour. Please find below a point-by-point list of answers to the reviewers’ concerns.

Reviewer #1

All my comments have been addressed but the revision must be done more carefully. Bibliography must be revised.

Response:

All references and respective numbers were carefully revised, as can be seen throughout the whole document. As part of this revision process, figure 3 also was revised.

For references 40, 116 and 119 the same paper has been cited. This paper does not correspond to what references 116 (page 683) and 119 (page 700) should deal with. The appropriate papers should be cited.

Response:

The references have been revised. Former reference 116 is now reference 118 and former reference 119 is now 121.

590-591: Figure 13 caption should be eliminated if figures 12 and 13 are not being included in the revised version. Figures 14 and 15 should be renumbered accordingly.

Response:

Thank you for your comment. Figures 14 and 15 were renumbered and former figure 13 caption was removed. Please see lines 604 and 608, 641 and 648, and 580, respectively.

1111: ref 125 does not display the correct authors names and order.

Response:

The authors names and order of this reference were revised. Please see reference 127.

In general, authors should revise English writing and correct some grammar and spelling mistakes

Response:

English grammar and spelling were revised. Please, see the corrections in green.
